# Trait variation in patchy landscapes: Morphology of spotted salamanders (*Ambystoma maculatum*) varies more within ponds than between ponds

Elizabeth T. Green[1,2], Anthony I. Dell[1,3], John A. Crawford[1], Elizabeth G. Biro[3,4], David R. Daversa 🔘 [1,3,5] *

1 National Great Rivers Research and Education Center (NGRREC), East Alton, IL, United States of America, 2 Department of Biology, University of North Carolina at Chapel Hill, Chapel Hill, NC, United States of America, 3 Department of Biology, Washington University in St. Louis, St. Louis, MO, United States of America, 4 Tyson Research Center, Washington University in St. Louis, St. Louis, MO, United States of America, 5 Institute of the Environment and Sustainability, La Kretz Center for California Conservation Science, University of California, Los Angeles, CA, United States of America

* ddaversa@gmail.com, ddaversa@ucla.edu

**Data Availability Statement:** The data and the R script used for the analyses and figures are publicly available on Figshare: https://figshare.com/articles/

## Abstract

The influence of intraspecific trait variation on species interactions makes trait-based approaches critical to understanding eco-evolutionary processes. Because species occupy habitats that are patchily distributed in space, species interactions are influenced not just by the degree of intraspecific trait variation but also the relative proportion of trait variation that occurs within- versus between-patches. Advancement in trait-based ecology hinges on understanding how trait variation is distributed within and between habitat patches across the landscape. We sampled larval spotted salamanders (*Ambystoma maculatum*) across six spatially discrete ponds to quantify within- and between-pond variation in mass, length, and various metrics associated with their relationship (scaling, body condition, shape). Across all traits, within-pond variation contributed more to total observed morphological variation than between-pond variation. Between-pond variation was not negligible, however, and explained 20–41% of total observed variation in measured traits. Between-pond variation was more pronounced in salamander tail morphology compared to head or body morphology, suggesting that pond-level factors more strongly influence tails than other body parts. We also observed differences in mass-length relationships across ponds, both in terms of scaling slopes and intercepts, though differences in the intercepts were much stronger. Preliminary evidence hinted that newly constructed ponds were a driver of the observed differences in mass-length relationships and morphometrics. General pond-level difference in salamander trait covariation suggest that allometric scaling of morphological traits is context dependent in patchy landscapes. Effects of pond age offer the hypothesis that habitat restoration through pond construction is a driver of variation in trait scaling, which managers may leverage to bolster trait diversity.

dataset/Amac_Morphology_R_scripts/24996476
DOI: https://doi.org/10.6084/m9.figshare.
24996476.v1.

**Funding:** This study was funded in part by the National Great Rivers Research and Education Center Grant/Award Number: NGRREC-IP2016-5. The funders had no role in study design, data collection and analysis, decision to publish, or preparation of the manuscript.

**Competing interests:** The authors have declared that no competing interests exist.

## Introduction

Morphological traits underpin many eco-evolutionary processes. The range of morphology exhibited by individuals of a species, a form of intraspecific trait variation [1], shapes the niche breadth of populations, which in turn affects their resiliency to environmental disturbances and biological invasions [2]. Because of this, models of population and community dynamics that incorporate intraspecific trait variation have become central to ecology and evolution [1,3]. A key outstanding challenge for this 'trait-based' paradigm is to understand landscape-level patterns in trait variation that explicitly consider trait differences at multiple scales [3,4].

Species tend to occur in landscapes comprising spatially discrete habitat patches, and at any given time, traits vary among individuals within patches and among groups between patches. Multiple forces drive within- and between-patch variation, with notable examples in vertebrates being predation risk [5–7], competitor presence [6], and phenology and ontogeny [8]. Regardless of the specific drivers of within and between-patch variation, theory predicts that the influence of trait variation on population and community dynamics depends on how variation is partitioned within versus between habitat patches [3,8–10]. Between-patch variation may arise through spatial differences in trait expression, spatial differences in species phenology or through stochastic factors leading to traits that are not necessarily adaptive [11]. In either case, between-patch variation potentially allows for a broader range of adaptive responses to environmental disturbances [3] and antagonistic interactions [2] than does within-patch variation. Alternatively, between-patch trait variation may heighten extinction risk by geographically isolating trait diversity to specific habitat patches. Population and community dynamics are influenced not just by the degree of intraspecific trait variation but also the relative proportion of trait variation that occurs within- versus between-patches over time [3,4].

Variation in how traits co-vary may also influence eco-evolutionary processes (but see [12]). Measures of trait co-variation (relationships between multiple individual traits) are useful for describing species growth patterns [13], their adaptive constraints [14], and complex morphological characteristics such as body shape [15]. Additionally, morphometrics integrates co-variation between body length, depth, and sometimes width, to characterize body shapes of individuals. Body shape, being a strong proxy of performance and fitness, is arguably a better predictor of species adaptive responses to environmental change than linear body measurements [15]. As such, shape and allometry can be powerful predictors of ecology and evolution in patchy landscapes.

In this study, we assessed within- and between-patch variation and co-variation in morphological traits in larval spotted salamanders (*Ambystoma maculatum*). Morphological variation in larval salamanders has been extensively studied, and the literature has identified multiple drivers of morphological variation in this group of animals, including predation risk [6,7], competition [5,6], geographic distance [16,17], and breeding phenology [17,18]. We build off this literature to quantify how morphological variation is partitioned within and between patches (i.e. ponds) in a patchy landscape. Specifically, we asked to what extent morphological trait variation in salamanders arises from individual trait differences within patches compared to group-level trait differences among patches. We also asked how consistent co-variation between morphological traits is across ponds. To address these questions, we sampled larval spotted salamanders among six spatially discrete ponds over the course of six weeks and measured body length and mass of over five hundred individuals. We used these data to quantify individual variation in mass and length within and among ponds during latter stages of larval development. We then examined the contribution of between-patch (i.e., between-pond) differences to the total observed variation and co-variation of those traits. Capitalizing on

differences in construction history of ponds at our study site, we also performed an examination of whether salamander mass, length, body condition, mass-length allometry, and shape were influenced by the age of ponds.

## Materials and methods

### Study species

Spotted salamanders are broadly distributed throughout the Northeastern and Midwestern United States [19]. Spotted salamanders are semi-terrestrial pond breeders, annually migrating from terrestrial hibernacula to reproduce in fishless wetlands, including constructed ponds [17,20] like those surveyed here (see below). Breeding in our study area occurs between March–April [21]. After hatching from eggs, aquatic larvae develop and metamorphose in 6–10 weeks [19]. Larvae feed on invertebrates and anuran tadpoles, and are themselves prey for adult salamanders and larger aquatic invertebrates such as odonate larvae and beetles [7]. Spotted salamander populations are useful systems to compare within- and between-patch morphological variation because: i) individuals occupy spatially discrete ponds [22]; and ii) larval stages exhibit substantial morphological plasticity in response to heterogeneity in biotic and abiotic conditions [7,23,24], which permits a range of trait expressions across individuals and ponds.

### Field sampling and husbandry

This study was executed with collection permits granted by the Missouri Department of Conservation and was approved by the Institutional Animal Care and Use Committee (IACUC) at the University of Illinois (#16203). We sampled ponds in east-central Missouri, distributed across three distinct conservation properties–Tyson Research Center (800 ha), Forest 44 Conservation Area (400 ha) and Shaw Nature Reserve (700 ha) (Fig 1). We focused on six ponds in which pilot surveys confirmed Spotted salamander larvae were present. Three ponds (Mincke Pond, Arthur Christ Pond, Beth's Pond) were constructed in 2008 for research purposes and had similar sizes and dimensions [25]. As part of a separate experiment, Rotenone was applied to Beth's Pond in 2008, which initially reduced microbial biodiversity [26], but the microorganismal community structure had returned to pre-treatment conditions well before sampling for this study [26]. The other three focal ponds (Salamander Pond, Forest 44 Pond, Shaw Pond) were older and variable in size (S1 File). Salamander Pond was created in 1965, and Forest 44 Pond and Shaw Pond between 1990 and 1996 (data extracted from Google Earth Historical Imagery). Mean water temperatures during the survey period were similar across ponds, ranging from 22–23 degrees Celsius (S1 File). Salamander density varied across ponds and ranged from 1–4 salamanders per square meter (S1 File). All ponds contained multiple predators of larval spotted salamanders including: dragonfly (*Anax* sp.) and damselfly (*Lestes* sp.) nymphs, diving beetles (*Dytiscidae* sp.), hyrdrophylid beetles (*Tropisternus* sp), and adult newts (*Notophthalmus viridescens*) (S1 File; E.G. Biro unpublished data). All ponds were located within forested habitats typical of temperate deciduous ecosystems found in the Midwestern US. A major highway bisected the study area, separating Forest 44 and Shaw ponds from the other four ponds.

We sampled each pond once, and one pond per week (30 June–08 August 2016), by dip-netting near the perimeter of the pond. We focused sampling on the latter stage of the developmental period of salamanders (Harrison stage 45–46; [27], when growth and development had slowed [28]. Because our sampling of ponds confounded space and time, between-patch morphological variation could have arisen through spatial heterogeneity of relevant factors (e.g. biotic and abiotic environment, genetics) or through temporal differences in salamander

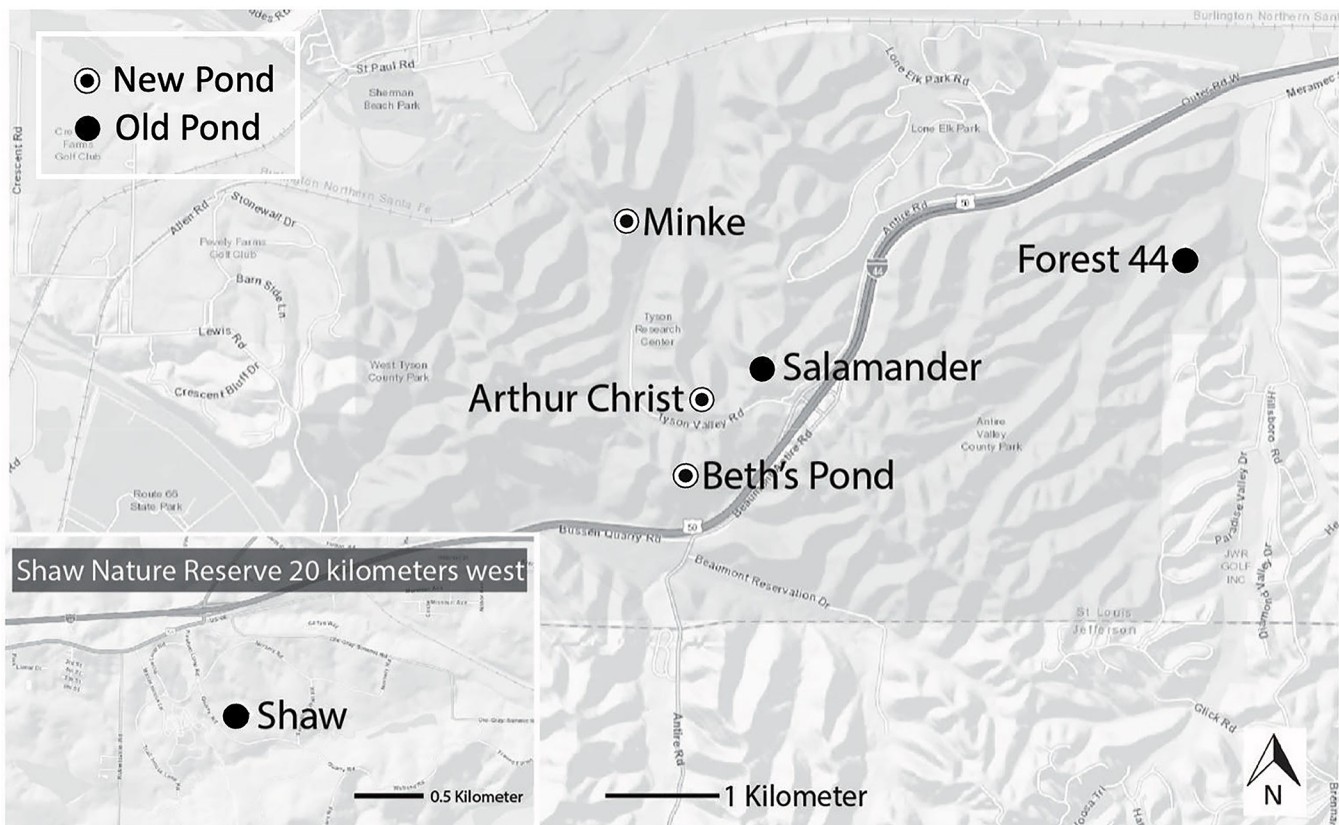

**Fig 1. Map of study area.** The six focal ponds were located in Eastern Missouri (US across three conservation areas. Mincke Pond, Arthur Christ Pond, and Beth's Pond are located in Tyson Research Center. Shaw Pond is located in the Shaw Nature Reserve. Forest 44 Pond is located in Forest 44 Conservation Area. All ponds occurred in Oak-Hickory forests typical of the region.

growth and ontogeny. We therefore make no inferences about the underlying drivers of observed variation. At each sampling event, we dip-netted until 10 minutes passed without a capture to maximize coverage of morphological variation within ponds. We retained all larvae that did not show overt signs of injury or illness (e.g., damaged tails or legs, tumorous growth, etc.) and immediately transported them to the National Great Rivers Research and Education Center (NGRREC)–less than 1 hr drive–where they were housed for seven days before being returned to their original ponds.

We housed larvae individually in circular plastic arenas (28 cm diameter) filled with 500 mL dechlorinated water (approximately 2.5 cm depth). Larvae were maintained at 18˚C with a 14:10 h light:dark cycle, consistent with ambient conditions at the surveyed ponds. Salamanders were fed a single gray tree frog (*Hyla chrysoscelis-versicolor*) tadpole on the fifth day as part of a separate experiment. Observing that not all salamanders ate the tree frogs fed to them the prior day, we omitted salamanders who ate tadpoles from data analysis to avoid biases in weight measurements.

### Trait measurements

On the sixth day after capture, we measured the length and mass of salamanders, distinguishing between head length, body length, and tail length. We photographed lateral and dorsal images of each larvae placed into clear tanks that minimized movement (S2 File). We blot-

dried individuals on paper towels and weighed them using a digital scale. We measured the length of salamander heads, bodies, and tails from images using ImageJ (S2 File) [29]. We then estimated body condition from length and mass measurements using the scaled mass index [30]. The scaled mass index calculates a body condition estimate using four variables: individual mass, individual length, population mean length, and the slope of the ordinary least squares regression of log(mass) on log(length) [30]. We used the total length measurements to calculate scaled mass index, which included tails, because larval salamander tails are used for fat deposition [31]. Further, mass correlated more strongly with total length than lengths of specific body segments (see Results), making total length a stronger metric for estimating body condition [30].

To obtain measurements of the shape of larvae we digitized landmarks on lateral images using the software *tpsDig2* [32]. Following [5] we tagged twenty landmarks that outlined larval shape (S2 File). Landmarks 1–3 described the shape of the head of the larvae, landmarks 4–11 described body shape, and landmarks 9–20 outlined tail shape. Landmarks were rotated, scaled by size, and aligned to a coordinated system using the Procrustes least-squares superimposition available in the *geomorph* package for R statistical software [33]. We conducted four principal component analyses to explore the scaled two-dimensional shape variation, again distinguishing head shape, body shape, tail shape and the overall shape, which combined heads, bodies and tails. The first and second principal component (PC) scores accounted for most of the variation in head shape (PC1: 37%, PC2: 27%), body shape (PC1: 33%, PC2: 25%), tail shape (PC1: 34%, PC2: 18%), and overall shape (PC1: 27%, PC2: 26%) (S2 File). We therefore used PC1 and PC2 as the shape metrics in our analyses.

## Data analyses

We calculated the coefficient of variation–the ratio of the standard deviation to the mean–as a standardized measure of individual morphological variation within ponds. We partitioned observed variation in salamander length, mass, and body condition to within- and between-pond trait differences with generalized linear mixed models (GLMM), using the *lme4* package in R [34]. Specifically, we calculated the intra-class correlation coefficient (ICC) from GLMMs that included 'pond' as a random intercept term [35]. The ICC, also called the variance partitioning coefficient [36], is the proportion of total variation in response variables that is attributable to group-level (between-pond in our case) differences. In our analysis, the ICC indicated the proportion of total observed variation in salamander length and mass that came from between-pond differences in those traits. We ran separate GLMMs for mass and the four length measurements–head length, body length, tail length, and total length. All models used a Gaussian error structure, as all morphological data were normally distributed. Our sample size did not permit reliable examination into the role of spatial autocorrelation in explaining between-pond variation.

We then assessed within- and between-pond variation in mass-length allometry. Specifically, we ran GLMMs to test whether the slopes and intercepts of mass-length regressions differed across the six focal ponds, again distinguishing head length, body length, and tail length. GLMMs included mass as the response, length as a fixed effect, pond as a random intercept term, and length as a random slope term. We scaled and log-transformed both length and mass and used a Gaussian error structure for the normalized data in all models. To test for differences in regression intercepts and slopes across focal ponds, we used likelihood ratio tests comparing the fit of models that included both terms with models omitting the random intercept or random slope term. We also calculated the marginal and conditional $R^2$ of the models using the *MUMIn* package in R. Marginal $R^2$ is a measure of the amount of variation in mass

that was explained by the fixed effect of length, while conditional $R^2$ considers variation explained by both fixed and random effect terms [37].

To further assess differences in the co-variation of morphological traits among focal ponds, we determined the extent to which between-pond variation in salamander head, body, tail, and overall (all segments combined) shape contributed to total observed variation in these multidimensional morphological traits. Again, we calculated the ICC from GLMMs, including 'pond' as a random intercept term. The shape data were also normally distributed, and so we used a Gaussian error structure for all models.

To capitalize on the stark dichotomy in the age of the focal ponds ('new' vs. 'old'), we assessed whether salamander mass, length, mass-length co-variation, and shape were influenced by pond age class. We used historical information described above to classify pond age as 'new' (N = 3; Mincke Pond, Arthur Christ Pond, Beth's Pond) or 'old' (N = 3; Salamander Pond, Forest 44 Pond, Shaw Pond) (S1 File). We ran GLMMs that included pond age (new vs. old) as a fixed effect, and the pond name as a random effect. To test how the age class of ponds influenced mass-length co-variation, we ran GLMMs with log-transformed mass as the response and log-transformed length, the focal factor (i.e., pond age) and their interaction as fixed effects. We also included salamander length (again, log-transformed) as a random slope term, and pond name as a random intercept term to account for possible differences in slopes and intercepts. We also ran GLMMs for our different length measures: head length, body length, tail length, and total length. We compared the fit of models including the pond age as a fixed effect with models omitting the pond age as a fixed effect, using likelihood ratio tests, to test the influence of pond age on the relationship between mass and length.

## Results

We measured a total of 519 spotted salamander larvae (Salamander pond: N = 65, Shaw: N = 88, Mincke: N = 119, Arthur Christ: N = 116 Forest 44: N = 101, Beth's: N = 30) and omitted 88 that ate tadpoles the day before measurement for a separate experiment. This omission left a total sample size for analysis of 429 salamanders (Salamander pond: N = 59, Shaw: N = 84, Mincke: N = 86, Arthur Christ: N = 87, Forest 44: N = 99, Beth's: N = 15). Within all ponds, salamander mass varied more among individuals than body condition and length measurements (Fig 2A, S1 File). Morphological variation was consistently lower in Beth's Pond (Fig 2B), though this may have been due to the lower sample size. Otherwise, there was no indication that specific ponds harboured most of the morphological variation (Fig 2A and 2B).

Between-pond differences in average trait values accounted for 20% to 41% of the total observed variation in salamander mass, body condition, and length measurements (Fig 3, Table 1). Between-pond differences accounted for proportionally similar amounts total variation in mass (41%), tail length (38%), and body length (36%). Between pond differences in head length (20%) and body condition (22%) contributed proportionally less to total observed variation than other traits (Table 1).

Salamander mass strongly co-varied with total length (Table 2). Mass also strongly co-varied with body length and tail length but was less correlated with head length (Table 2). There were detectable differences in the intercepts of mass-length relationships across ponds (including pond as a random intercept term improved model fit; mass-head length: $X^2_1 = 110.15$, p < 0.001; mass-body length: $X^2_1 = 131.91$, p < 0.001; mass-tail length: $X^2_1 = 39.50$, p < 0.001; total length: $X^2_1 = 70.76$, p < 0.001; Fig 4). For a given length, individuals from Salamander Pond tended to be heavier than individuals from other ponds (Fig 4) whereas individuals from Beth's Pond were generally lighter in mass per unit length (Fig 4). The slopes of mass-length relationships were also influenced by pond of capture, except for mass-body length

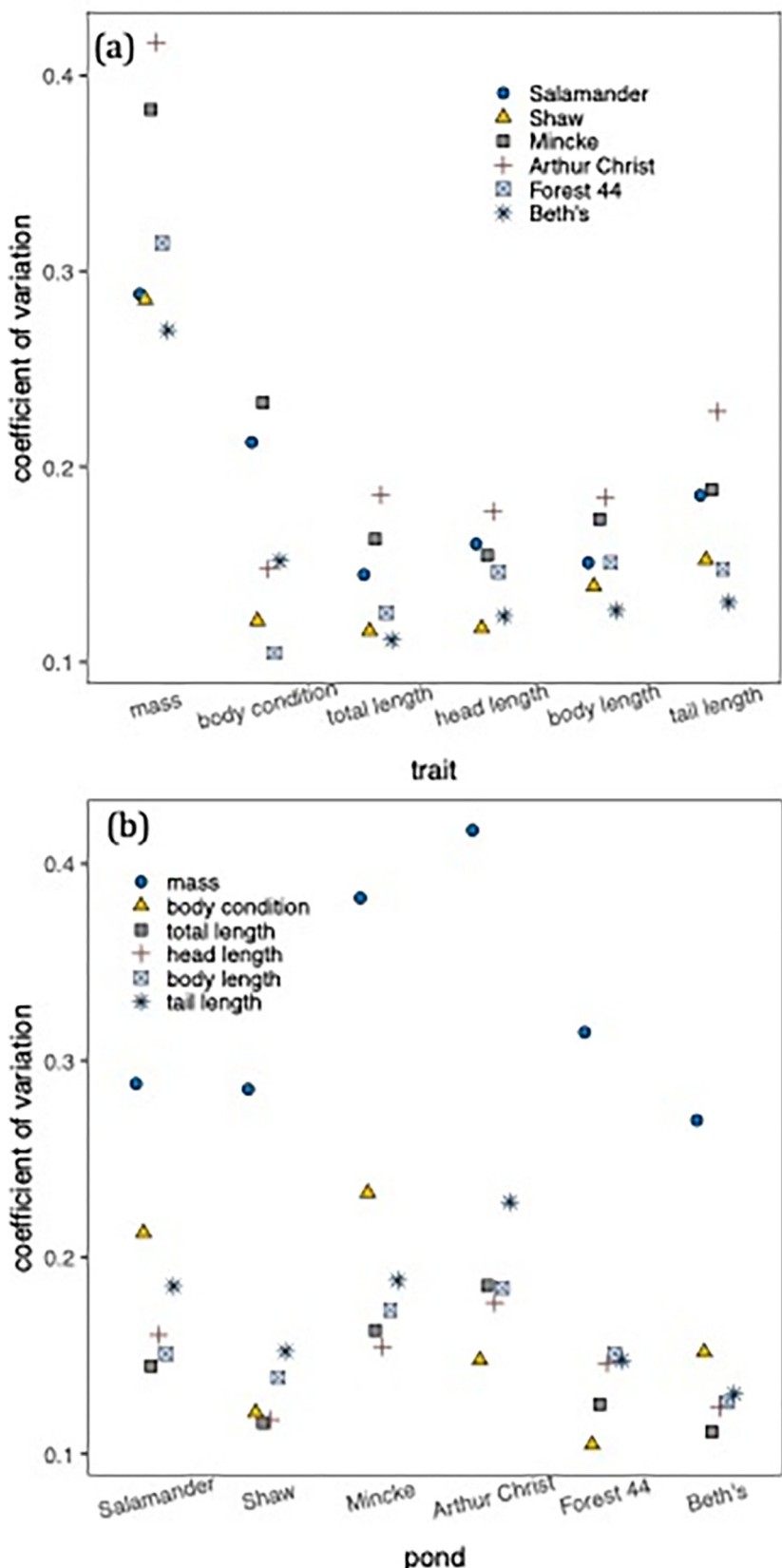

Fig 2. Within-pond variation in salamander morphology. The coefficient of variation, or the extent of variance in relation to mean trait values, is shown for **(a)** mass and length measures and **(b)** the six ponds where we sampled salamanders, ordered from left to right in the chronological order in which they were sampled. Color-symbol combinations distinguish pond of capture in (a) and the focal trait in (b). Note that the same data are reported in (a) and (b).

relationships (mass-head length: $X^2_2 = 6.54$, p = 0.038; mass-body length: $X^2_2 = 3.11$, p = 0.211; mass-tail length: $X^2_2 = 14.21$, p = 0.001; mass-total length: $X^2_2 = 6.95$, p = 0.031; Table 2, Fig 4). Slopes of mass-tail length scaling differed most across ponds (Fig 4). However, pond-level differences in scaling slopes were generally weaker than pond-level differences in scaling intercepts (Fig 4). The slope exponents were always $< 3$ (S1 File), indicating that larger salamander larvae generally had more elongate heads, bodies, and tails than smaller larvae.

In terms of morphometrics, two principal components consistently explained greater than 10% of the variation (PC1–27–37%; PC2–18–27%; S2 File) and were used in the analyses. For the overall shape (combining heads, bodies, and tails), PC1 and PC2 values explained an almost equal amount of the variation (PC1: 27%, PC2: 26%) illustrating that there is similar variation in overall length and depth of larval shape. For the shapes of specific body segments (head, body, or tail), PC1 values corresponded to length, while PC2 represented shape depth. PC1 and PC2 values increased with elongation of shape and increasing length: height ratio. Between-pond trait differences contributed 25% of the total observed variation in tail shape, compared with 10%, 9%, and 10% of head, body, and overall shape, respectively (Table 1). Furthermore, there was little evidence that PC scores were clustered by pond (Fig 5), indicating a weak signal of between-pond variation in salamander shape.

Additionally, pond age did not influence salamander mass or any measures of length (S3 File). However, pond age influenced the scaling of mass with head and tail lengths (S3 File) as well as the overall shape of salamanders (S3 File).

## Discussion

Measuring the lengths and masses of salamander larvae across spatially discrete ponds showed that the majority of morphological variation and co-variation occurred within ponds. The substantial within-pond variation in salamander morphology emphasizes the pronounced morphological plasticity these animals exhibit and suggests that local factors, such as microhabitat heterogeneity, influence salamander morphology. Spatial connectivity may also have been involved in the observed partitioning of morphological variation. Given that four of the six ponds were spaced within the documented dispersal ranges of salamanders [16,22], movement between ponds could have reduced the contribution of between-pond differences to morphological variation by sustaining mixing of genotypes and phenotypes. At the landscape scale, local and spatial factors likely interact to shape the within- and between-pond partitioning of morphological variation that we observed in salamanders.

Between-pond morphological differences were not negligible. Between 20%-41% of total observed variation was attributable to between-pond differences, depending on the specific morphological trait. Again, salamander mass exhibited more between-pond variation than other morphological traits. Interestingly, this did not translate into equally strong between-pond variation in body condition estimates based on mass-length scaling. Because body condition is arguably more closely linked to amphibian fitness than length or mass measures [31,38], the relatively strong between-pond variation in salamander mass is not necessarily indicative of pond-level differences in salamander fitness. Our sampling could not distinguish between spatial and temporal variation in salamander morphology, and so between-patch variation could have arisen either through environmental/genetic differences among ponds,

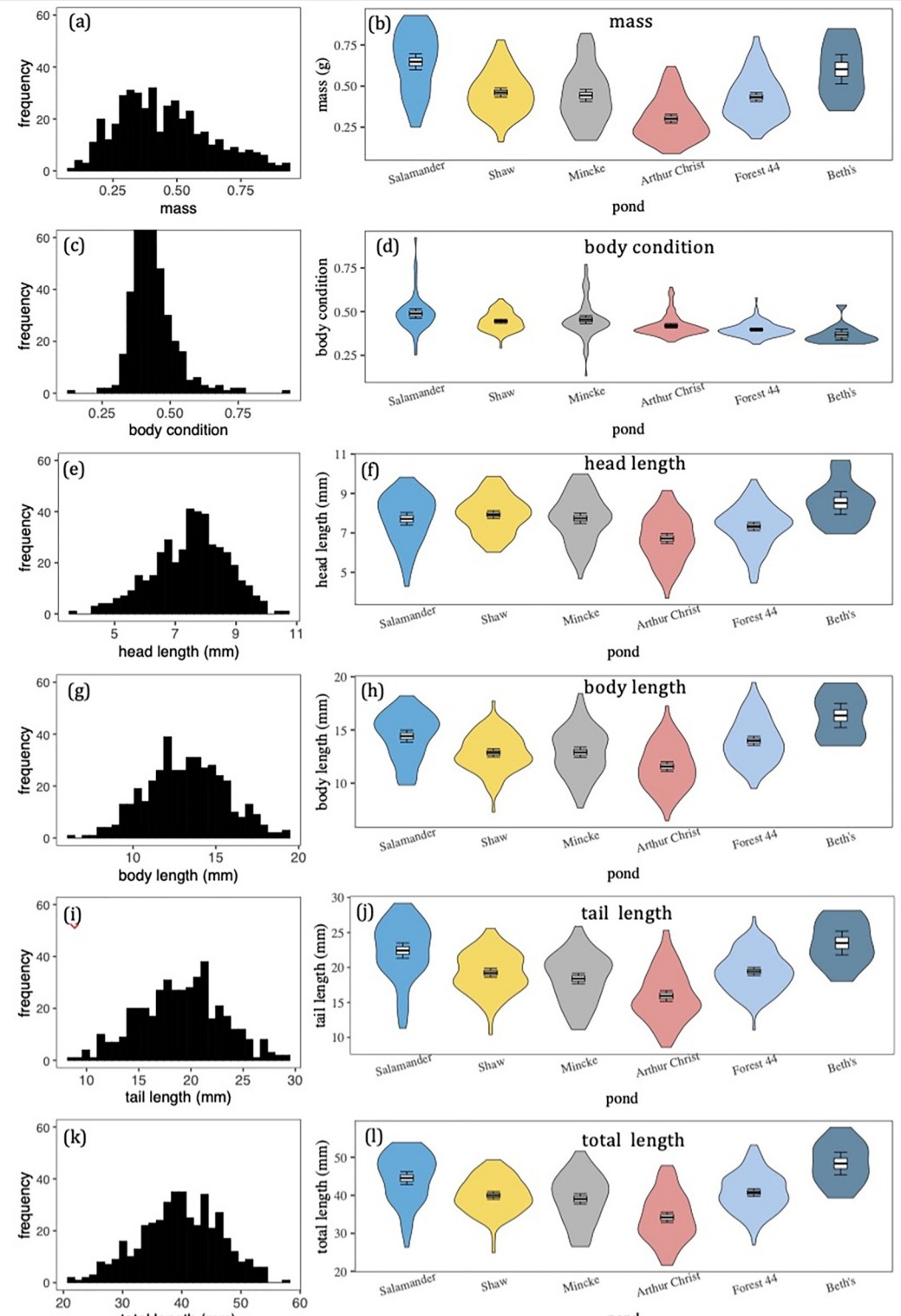

**Fig 3. Salamander mass and length variation within and between ponds.** Panels on the left (**a,c,e,g,i,k**) are frequency distributions of mass, body condition and length measurements across the six focal ponds. Violin plots on the right-side panels (**b,d,f,h,j,l**) visualize within- and between-pond morphological variation. Box plots within the violin plots denote the mean, standard error, and 95% confidence intervals of trait measures. Ponds in the right-side panels are ordered from left to right in the chronological order in which they were sampled.

**Table 1. Proportion of mass, length, and shape variation attributable to between-pond differences.**

| body segment | pond-level variance | SD | residual variance | SD | ICC (%) | 95% CI—lower | 95% CI -upper |
|---|---|---|---|---|---|---|---|
| | | | **mass & body condition** | | | | |
| mass | 0.015 | 0.123 | 0.022 | 0.149 | 40.6 | 8.11 | 62.83 |
| condition | 0.002 | 0.040 | 0.006 | 0.075 | 22.3 | 3.49 | 41.34 |
| | | | **length** | | | | |
| head | 0.32 | 0.56 | 1.25 | 1.12 | 19.9 | 3.24 | 40.68 |
| body | 2.46 | 1.57 | 4.35 | 2.08 | 36.0 | 8.38 | 60.42 |
| tail | 7.10 | 2.67 | 11.34 | 3.37 | 38.5 | 8.67 | 61.87 |
| combined | 21.90 | 4.68 | 33.15 | 5.76 | 39.8 | 8.85 | 62.00 |
| | | | **shape—PC1** | | | | |
| head | 0.0004 | 0.0197 | 0.0029 | 0.0537 | 11.80 | 0.67 | 26.93 |
| body | 0.0002 | 0.0142 | 0.0023 | 0.0476 | 8.20 | 0.00 | 20.26 |
| tail | 0.0005 | 0.0234 | 0.0024 | 0.0487 | 18.70 | 4.56 | 46.42 |
| combined | 0.0001 | 0.0087 | 0.0010 | 0.0324 | 6.70 | 0.07 | 19.99 |
| | | | **shape—PC2** | | | | |
| head | 0.0001 | 0.0099 | 0.0029 | 0.0468 | 4.30 | 0.00 | 12.34 |
| body | 0.0002 | 0.0139 | 0.0016 | 0.0404 | 10.50 | 0.57 | 24.26 |
| tail | 0.0003 | 0.0173 | 0.0013 | 0.0358 | 18.90 | 3.10 | 37.87 |
| combined | 0.0001 | 0.0122 | 0.0010 | 0.0316 | 13.00 | 1.27 | 30.22 |

Displayed are the variance components of generalized linear mixed models used for our analyses of within- and between-pond variation in salamander mass, length, and two primary principal components describing shape (PC1 & PC2). The intra-class correlation coefficient (ICC), or the variance partitioning component, is the proportion of between-pond variation explaining total observed trait variation (between-pond variance + residual variance). Higher ICC values denote higher degree of between-pond variation in salamander traits. CI = confidence intervals of the ICC derived from bootstrapping over 500 resampling events.

differences in salamander phenology among ponds, or within-individual growth and development throughout the survey period. Although we sampled salamander larvae during latter developmental stages, evidenced by all salamanders being in the final Harrison stages (45–46) [27], substantial growth and development in these latter stages can occur and may have contributed to some of the observed variation. We did not observe monotonic increases in body lengths and mass throughout the sampling period, which should have been the case if between-pond morphological variation were driven completely by within-individual growth and development. We hypothesize that the observed between-pond variation arose from a mix of pond-level differences in environmental conditions and salamander genetics as well as

**Table 2. Variation in intercepts and slopes of mass-length relationships.**

| body segment | pond-level intercept variance | SD | pond-level slope variance | SD | residual variance | SD | marginal R2 | conditional R2 |
|---|---|---|---|---|---|---|---|---|
| | | | | **mass-length co-variation** | | | | |
| head | 0.095 | 0.308 | 0.062 | 0.250 | 0.016 | 0.128 | 0.25 | 0.50 |
| body | 0.037 | 0.191 | 0.013 | 0.113 | 0.008 | 0.088 | 0.63 | 0.76 |
| tail | 0.103 | 0.322 | 0.048 | 0.219 | 0.007 | 0.082 | 0.72 | 0.79 |
| combined | 0.107 | 0.328 | 0.032 | 0.178 | 0.005 | 0.068 | 0.80 | 0.86 |

Displayed are the variance components of generalized linear mixed models used for our analyses of between-pond variation in the scaling of salamander mass with different length measurements. Models included length (log-transformed) as a fixed effect and a random slope term, and pond as a random intercept term. Marginal $R^2$ denotes the amount of variation in mass explained by the length measurement alone, whereas conditional $R^2$ considers variation explained by the random intercept and slope terms.

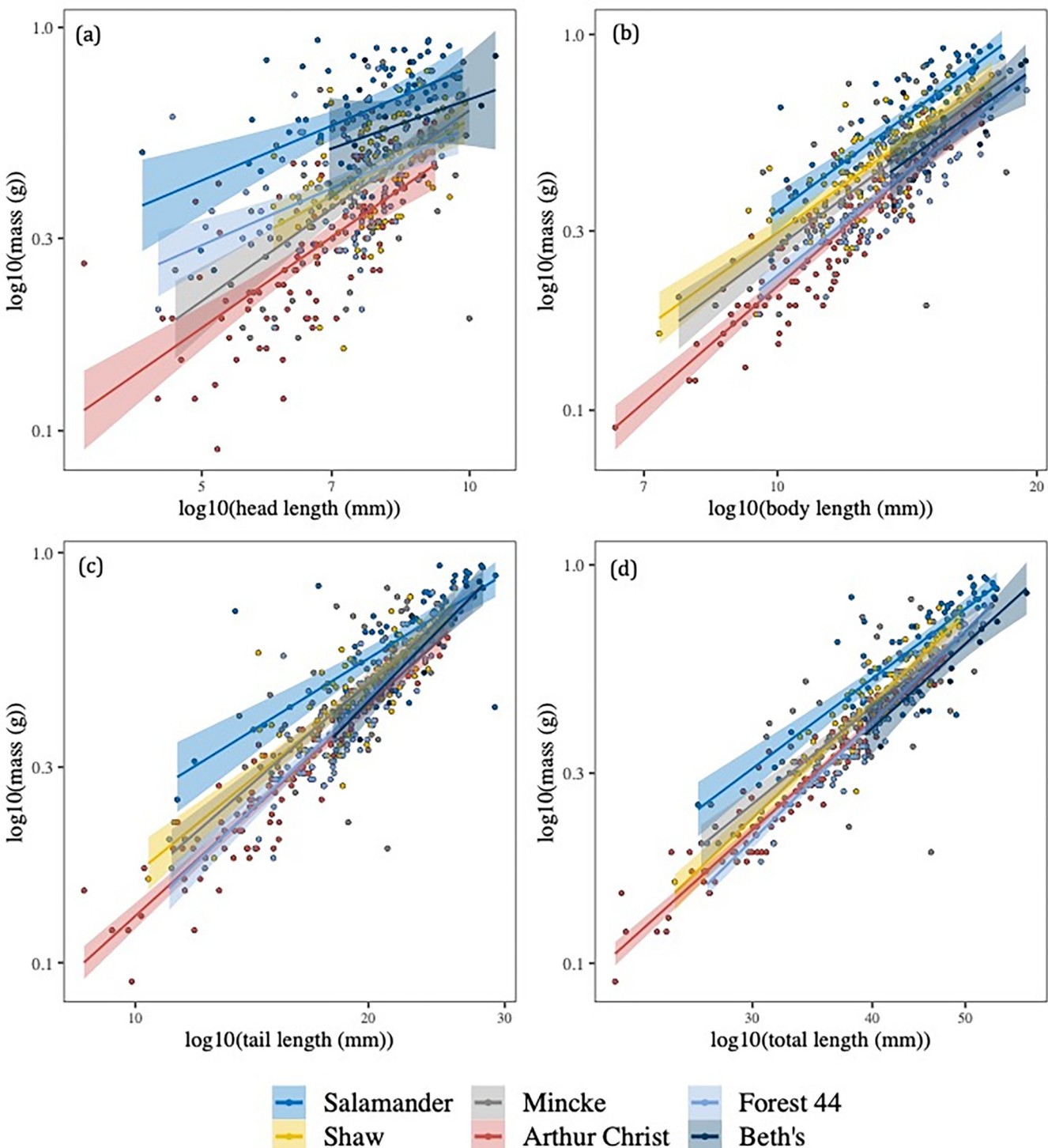

**Fig 4. Between-pond variation in mass-length allometry in salamanders.** Regression lines of salamander mass with (a) head length, (b) body length, (c) tail length, and (d) are shown for the six focal ponds, distinguished by line colors. Shaded areas show the 95% confidence intervals of the regression lines. Mass and length are plotted on a $\log_{10}$ scale in all cases.

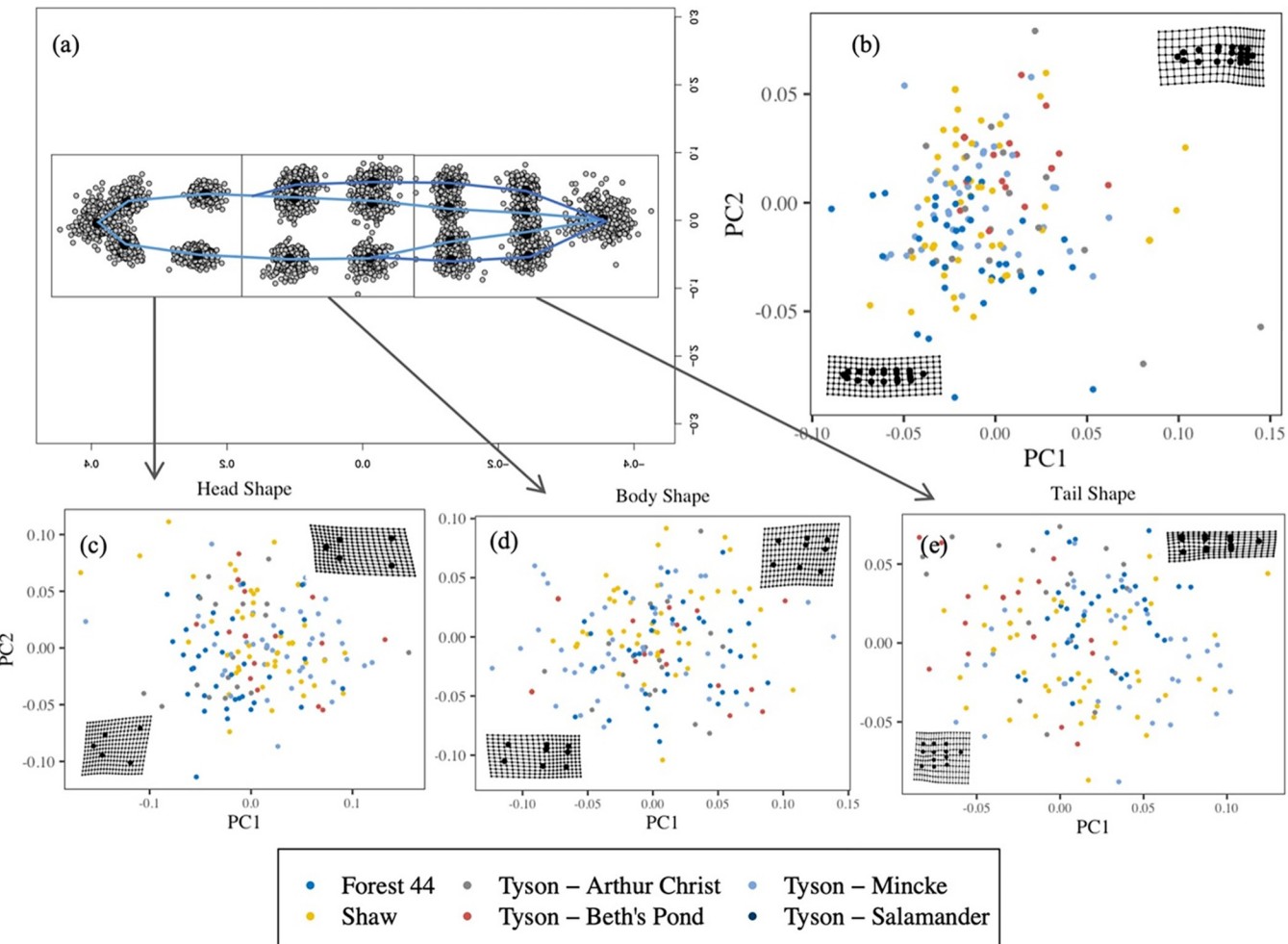

**Fig 5. Salamander shape variation within and among ponds.** The shape values are based on sets of landmarks at different points along the lateral surface of salamander bodies (a). The overall (b), head (c), body (d), and tail (e) shape of salamanders collected from the six focal ponds. The shape values are based on sets of landmarks at different points along the lateral surface of salamander bodies. PC1 and PC2 values increase with elongation of shape and increasing length: height ratio.

pond-level asynchronies in salamander ontogeny. Previous studies point to variable environmental conditions, namely predation risk and conspecific density, as key drivers of between-pond variation in larval salamander morphology [5,6,39,40]. Multiple predators of larval salamanders were found in our focal ponds (S1 File), and conspecific density also varied between ponds (S1 File). Predation risk and conspecific density could therefore have played a role in driving the between-pond morphological variation observed here, though we could not test for this. Our analysis highlighted pond age as an additional, and less documented, putative environmental variable that may be involved. The distinct dichotomy among our focal ponds in their age and history appear to have influenced allometric relationships between salamander length and mass as well as the overall shape of salamanders. The new ponds were generally smaller, shallower, and had artificial lining, which may have influenced trait co-variation. While we refrain from drawing firm conclusions from these results because of our small sample size (3 'new' ponds and 3 'old' ponds), the observed morphological differences between salamanders from 'new' vs. 'old' ponds offer an intriguing hypothesis that constructed ponds increase morphological diversity. The question remains as to the specific environmental,

genetic and ontogenetic explanations for between-pond variation in larval salamander morphology observed here. Regardless, our findings indicate that a non-negligible proportion of morphological variation in late-stage salamander larvae arises through pond-level trait differences.

In terms of co-variation of morphological traits, between-pond differences were particularly evident in salamander tails, consistent with previous studies [5,6,39]. Scaling of mass with salamander tail lengths differed across ponds both in terms of the intercepts and the slopes of the relationships, and between-pond differences contributed more to total observed variation in tail shape than for head or body shape. Some tail variation may be attributable to tail damage, and though we did not observe tail damage in any captured individuals, salamanders quickly regenerate tails following damage. Lasting effects on tail morphology may occur from tail damage as well [41]. Again, previous work argues that predator- and competitor-driven phenotypic plasticity are key drivers of variation in larval salamander tail morphology [5,6,39]. Exposure to invertebrate predators elicits growth of longer tail muscles and deeper fins [5,39], whereas exposure to dense conspecific competition results in smaller tails [6]. Tail morphology also influences swimming performance, and being meso-predators, swimming performance is critical to capturing prey and outcompeting conspecifics [40]. Tails may therefore be closely linked to fitness, and under selection, from a performance standpoint [42].

Between-pond variation in the allometric relationship between salamander mass and length measurements was evident both in scaling intercepts and slopes, with the exception that the scaling of mass with body length remained consistent across ponds. The presence of variation in mass-length scaling suggests that allometric patterns are context dependent, varying across the landscape. Although slopes of allometric relationships varied across ponds, the variation was much weaker than that observed with intercepts. This pattern of allometric scaling, in which intercepts of trait relationships differ more strongly than slopes, is consistent with allometric relationships documented across many other taxa [14], suggesting a general constraint to the plasticity and evolution of the slopes of trait relationships.

The inclusion of morphological diversity data in biodiversity conservation stems from the idea that different populations of the same species are not equal in terms of eco-evolutionary history. As such, exploring various approaches to the conservation of morphological diversity is important to developing strategies for reducing biodiversity losses under global change [43]. The mix of within- and between-pond morphological variation in salamanders provides promise that pond construction can utilize local and regional processes to bolster morphological diversity. Capitalizing on the presence of new constructed ponds in our study area, we made a preliminary comparison of salamander morphology and allometry between new and old ponds. Although our analysis did not detect differences in mass or length of salamanders between new and old ponds, we did find differences in mass-length relationships and morphometrics (S3 File). The reasons behind these differences could not be determined and should be examined further. Regardless of the specific reasoning, this preliminary finding offers an exciting hypothesis that habitat restoration through pond construction drives variation in trait scaling, which managers may leverage to bolster trait diversity.

## Supporting information

**S1 File. Further details on focal ponds and statistical analyses.** Supporting information for the manuscript contains additional information on pond characteristics, tables reporting outputs of statistical models, and data on the invertebrates.
(DOCX)

**S2 File. Salamander morphology.** A visualization of the landmarks used for the morphometric characterizations as well as a table reporting the variance explained by principle components is shown.
(DOCX)

**S3 File. Effects of pond age on salamander morphology.** Detailed methods and results on the analysis of the extent to which pond age influence salamander morphology. The file includes one table reporting model outputs and two figures visualizing the observed patterns.
(DOCX)

## Acknowledgments

We thank members of the Tyson Research Station for their support of our field sampling, J. Grady for assistance with data analyses and visualization, J. Messier for assistance with the statistical analyses, and S. Trombulak for assistance with data analyses and comments that significantly improved this manuscript. None of the authors experienced conflict of interest that could have influenced the objectivity of this study.

## Author Contributions

**Conceptualization:** David R. Daversa.

**Data curation:** Elizabeth G. Biro, David R. Daversa.

**Formal analysis:** Elizabeth T. Green, John A. Crawford, David R. Daversa.

**Investigation:** Elizabeth T. Green, Anthony I. Dell, Elizabeth G. Biro, David R. Daversa.

**Methodology:** Elizabeth T. Green, Anthony I. Dell, David R. Daversa.

**Project administration:** David R. Daversa.

**Supervision:** Anthony I. Dell, John A. Crawford, David R. Daversa.

**Validation:** David R. Daversa.

**Visualization:** David R. Daversa.

**Writing – original draft:** Elizabeth T. Green, David R. Daversa.

**Writing – review & editing:** Elizabeth T. Green, Anthony I. Dell, John A. Crawford, Elizabeth G. Biro, David R. Daversa.

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
