## [Decision Letter · Decision Letter 0]

18 Dec 2023

PONE-D-23-36372Trait variation in patchy landscapes: morphology of spotted salamanders (Ambystoma maculatum) varies more within ponds than between pondsPLOS ONE

Dear Dr. Daversa,

Thank you for submitting your manuscript to PLOS ONE. After careful consideration, we feel that it has merit but does not fully meet PLOS ONE’s publication criteria as it currently stands. Therefore, we invite you to submit a revised version of the manuscript that addresses the points raised during the review process. Please submit your revised manuscript by Feb 01 2024 11:59PM. If you will need more time than this to complete your revisions, please reply to this message or contact the journal office at plosone@plos.org. Please include the following items when submitting your revised manuscript:A rebuttal letter that responds to each point raised by the academic editor and reviewer(s). You should upload this letter as a separate file labeled 'Response to Reviewers'.A marked-up copy of your manuscript that highlights changes made to the original version. You should upload this as a separate file labeled 'Revised Manuscript with Track Changes'.An unmarked version of your revised paper without tracked changes. You should upload this as a separate file labeled 'Manuscript'.If applicable, we recommend that you deposit your laboratory protocols in protocols.io to enhance the reproducibility of your results. Protocols.io assigns your protocol its own identifier (DOI) so that it can be cited independently in the future. For instructions see: https://journals.plos.org/plosone/s/submission-guidelines#loc-laboratory-protocols. Additionally, PLOS ONE offers an option for publishing peer-reviewed Lab Protocol articles, which describe protocols hosted on protocols.io. Read more information on sharing protocols at https://plos.org/protocols?utm_medium=editorial-email&utm_source=authorletters&utm_campaign=protocols.

We look forward to receiving your revised manuscript.

Kind regards,

Dr. Janice L. Bossart

Academic Editor

PLOS ONE

Journal Requirements:

"We thank members of the Tyson Research Station for their support of our field sampling, J.

Grady for assistance with data analyses and visualization, J. Messier for assistance with the 

statistical analyses, and S. Trombulak for assistance with data analyses and comments that 

significantly improved this manuscript. This project was conducted in accordance with 

University of Illinois IACUC #16203 and funded, in part, by the NGRREC intern program. 

None of the authors experienced conflict of interest that could have influenced the 

objectivity of this study"

"The author(s) received no specific for this work."

5. Please note that your Data Availability Statement is currently missing the DOI/accession number of each dataset. If your manuscript is accepted for publication, you will be asked to provide these details on a very short timeline. We therefore suggest that you provide this information now, though we will not hold up the peer review process if you are unable.

7. We note that you have referenced (Tables S1 and S4; E.G. Biro unpublished data) which has currently not yet been accepted for publication. Please remove this from your References and amend this to state in the body of your manuscript: (Tables S1 and S4; E.G. Biro unpublished data) as detailed online in our guide for authors

Additional Editor Comments:

Your manuscript was favorably received by both reviewers, with both indicating only minor revisions were necessary.  Please carefully consider two significant points they raised regarding analysis and interpretation of between-pond variation:  Consider whether you can use date as a covariate in your models as an effective way to control for temporal effects, which might then allow you to explicitly explore/include the role of environmental variables.  Another important consideration is whether spatial autocorrelation exists among ponds or not, and how that might relate to factors contributing to between-pond variation. Some additional minor problems I noted when reading through your manuscript.Lines 62-63.  This is very vertebrate focused text, given the preceding sentence is inclusive of all species, and generally not reflective of what might drive within and between patch variation in, e.g. herbivorous insects, where major drivers are the specific array of host species present and the relative quality and abundance of each.Line 297-298.  Wording issue.  Word(s) missing?Line 329.  'driving to' should be 'driving the'Line 336-338.  Wording issue.  Do you mean '...remains as to the specific...' ?Line 377.  Change 'reasoning' to either 'reasons' or 'causes'

Reviewers' comments:

Reviewer's Responses to Questions

**Comments to the Author**

1. Is the manuscript technically sound, and do the data support the conclusions?

Reviewer #1: Yes

Reviewer #2: Yes

2. Has the statistical analysis been performed appropriately and rigorously? 

Reviewer #1: Yes

Reviewer #2: Yes

3. Have the authors made all data underlying the findings in their manuscript fully available?

Reviewer #1: Yes

Reviewer #2: No

4. Is the manuscript presented in an intelligible fashion and written in standard English?

Reviewer #1: Yes

Reviewer #2: Yes

5. Review Comments to the Author

Reviewer #1: The authors present the results of an investigation of morphological variation among salamanders in six ponds that is motivated by 1) the recognition that individual variation can have important ecological implications and, in a more applied context, 2) a question about whether artificial ponds that might be used in restoration might create trait variation or change the scaling of traits (i.e. allometric relationships). The study is limited to the questions addressed because, as the authors explain, geography and time were confounded such that explorations of mechanisms could not be attempted. Thus, the study is a detailed exploration of trait variance that partitions within- and between - pond effects for trait variation and allometric variation. Given the potential importance of individual variation, this project provides some insight into how trait variation is organized in these salamanders.

The authors repeatedly refer to this as preliminary. I object to this language because it has negative connotations that perhaps highlight the limitations of the sampling. Perhaps that conveys the authors' assessment, but it distracts from the main point as written which is the description of individual variation.

With respect to the limitations, would it be possible to simply use day of the year as a covariate to control for sampling date differences? This would assume that life history events are synchronized across the six ponds, but would facilitate a deeper exploration of environmental variables.

minor comments:

line 60: change "comprised of" to "comprising"

lines 66-67, 76, 78, 126, 144, 179, 184: references here don't conform to the rest of the ms formatting

lines 93-95: I wonder if the context for the study would be more clear if this question included the idea that the artificial ponds differing in age. That is one of the interesting results.

lines 186-190: One could also use MacArthur's broken stick approach, or a similar inertia method, to assess how many PC axes are sufficient.

line 204: I'm not sure why Figure 2 is referenced for this sentence about the data being normally distributed.

line 211: I think this needs more explanation. How does this multiplication 1) facilitate convergence and, 2) "standardize the units with length measurements"? For example, why not center and scale mass and length?

line 329: change "diving to" to "driving the"

Reviewer #2: I enjoyed reading this manuscript. It’s well-written, and the motivation was well justified and explained, as were the methods. I particularly appreciated the explanations given for modelling choices throughout the methods section. I have no major concerns, only a minor comment:

It seems the authors were careful to minimize the possibility that growth differences and temporal sampling contributed to major between-pond variation. I wonder if the authors looked at spatial autocorrelation among the sample sites. From the results it doesn’t seem obvious that there was spatial autocorrelation or potential for isolation by distance patterns in trait variation (with one site being markedly further away). But perhaps a lack of autocorrelation would also be interesting and support the idea that trait variation is more related to the pond-specific properties of salamander populations and environmental factors. In any case whether spatial autocorrelation was explored, or why the authors chose not to, should be noted in the Methods.

L234: predator density? Is this supposed to be here?

Fig S3: the confidence intervals are either hard to see or missing

6. PLOS authors have the option to publish the peer review history of their article (what does this mean?). If published, this will include your full peer review and any attached files.

Reviewer #1: No

Reviewer #2: No

---

## [Author Response · Author response to Decision Letter 0]

30 Jan 2024

Dear editors, reviewers and journal staff: 

Thank you for taking the time to review our manuscript and for the constructive feedback to improve the work. We have addressed the line edits in our revision, which you can see in the track revised manuscript with track changes. We have also revised the title page to match the PLOS ONE style requirements and removed the funding information from the Acknowledgements section. Our revised funding statement is below –

“This study was funded in part by the National Great Rivers Research and Education Center Grant/Award Number: NGRREC-IP2016-5”

We have also published our raw data and R scripts to the Figshare Data Repository. We have embargoed the raw data, and we will release the embargo upon acceptance of the manuscript. The R script for the data analyses and figures is available at: https://figshare.com/articles/dataset/Amac_Morphology_R_scripts/24996476

DOI: https://doi.org/10.6084/m9.figshare.24996476.v1

We provide responses to each of the review comments below: 

Additional Editor Comments:

Your manuscript was favorably received by both reviewers, with both indicating only minor revisions were necessary. Please carefully consider two significant points they raised regarding analysis and interpretation of between-pond variation: Consider whether you can use date as a covariate in your models as an effective way to control for temporal effects, which might then allow you to explicitly explore/include the role of environmental variables. Another important consideration is whether spatial autocorrelation exists among ponds or not, and how that might relate to factors contributing to between-pond variation.

Response: Thanks for taking the time to consider our manuscript. We have considered the two main comments pertaining to (a) the addition of day-of-year as a covariate in our statistical models and (b) the possibility of autocorrelation. Adding the day-of-year covariate did not affect model performance because the factor varied perfectly with our random effect – the ID of ponds. This is due to our sampling design, where we surveyed one pond/week, and a different pond each week; pond ID changed perfectly in tandem with day-of-year. In terms of spatial autocorrelation, we decided against a formal analysis because the results would have been unreliable with our small sample size of six ponds. Judging from the patterns of trait partitioning among ponds, we are doubtful that spatial autocorrelation was a factor. However, we do not have the breadth of data and sampling locations to reliably test this hypothesis, and we now state so in the manuscript (Line 244). Below, we elaborate on how we came to these decisions. Although in the end no changes to the analyses were made, we assure you that ample time was given to consider the options. 

Some additional minor problems I noted when reading through your manuscript.

Lines 62-63. This is very vertebrate focused text, given the preceding sentence is inclusive of all species, and generally not reflective of what might drive within and between patch variation in, e.g. herbivorous insects, where major drivers are the specific array of host species present and the relative quality and abundance of each.

Response: This is a good point. We have clarified the scope of this statement by adding ‘in vertebrates’ after ‘some notable examples...’ (Line 76). We considered adding examples pertaining to invertabrates, but because our study focuses on a vertebrate species, we decided for brevity to stick to vertebrate examples. 

Line 297-298. Wording issue. Word(s) missing?

Response: Yes, the sentence was missing the word ‘in’ after ‘involved. We have corrected this error: “Spatial connectivity may also have been involved in the observed partitioning of morphological variation” (Line 366). 

Line 329. 'driving to' should be 'driving the'

Response: Changed (Line 403) 

Line 336-338. Wording issue. Do you mean '...remains as to the specific...' ?

Response: Yes, and changed (Line 412) 

Line 377. Change 'reasoning' to either 'reasons' or 'causes'

Response: we changed to ‘reasons’ (Line 462) 

Reviewer #1:

“The authors present the results of an investigation of morphological variation among salamanders in six ponds that is motivated by 1) the recognition that individual variation can have important ecological implications and, in a more applied context, 2) a question about whether artificial ponds that might be used in restoration might create trait variation or change the scaling of traits (i.e. allometric relationships). The study is limited to the questions addressed because, as the authors explain, geography and time were confounded such that explorations of mechanisms could not be attempted. Thus, the study is a detailed exploration of trait variance that partitions within- and between - pond effects for trait variation and allometric variation. Given the potential importance of individual variation, this project provides some insight into how trait variation is organized in these salamanders.”

Response: We are glad to hear that the reviewer finds our project important despite the noted limitations.

“The authors repeatedly refer to this as preliminary. I object to this language because it has negative connotations that perhaps highlight the limitations of the sampling. Perhaps that conveys the authors' assessment, but it distracts from the main point as written which is the description of individual variation.”

Response: We referred to the analysis between “new” and “old” ponds as preliminary because of the low replication of ponds (N=6), however, we understand your objection and the negative connotation. To discuss our analysis, without overselling the statistical power, we have removed the “preliminary” language from the methods and results and instead describe the limitations in the discussion section, which we feel is appropriate and warranted.

“With respect to the limitations, would it be possible to simply use day of the year as a covariate to control for sampling date differences? This would assume that life history events are synchronized across the six ponds, but would facilitate a deeper exploration of environmental variables.”

Response: We could not include day of year as a covariate because of its collinearity with the pond IDs. We sampled one pond/week, and a different pond each week. With this design, the day of year varies equally with the pond identification where we sampled. Because our models already include ‘pond ID’ as a random effect, adding day of year would not add further explanatory power. Just as a check of this, we went back to a subset of our models and added the day of year as a covariate. The result was that the day of year variable was automatically dropped due to singularity, meaning essentially that the two variables vary equally. 

minor comments:

lines 66-67, 76, 78, 126, 144, 179, 184: references here don't conform to the rest of the ms formatting

Response: We updated the references so that they now follow the numbered format used by this journal. Thanks for pointing this out. 

lines 186-190: One could also use MacArthur's broken stick approach, or a similar inertia method, to assess how many PC axes are sufficient.

Response: In response to your suggestion, we ran a broken stick model on the decomposed principal component values. For the whole body shape the cut off was found to be the third component, while for the head, body, and tail we confirmed the cutoff at the second component. Since the cutoff was found to be the third component in only one of the shape metrics, we kept our previously decided cutoff at the second PC axis to allow for comparison between shapes.

line 211: I think this needs more explanation. How does this multiplication 1) facilitate convergence and, 2) "standardize the units with length measurements"? For example, why not center and scale mass and length?

Response: We had originally explored multiplication of the response variables as a technique for getting the mixed effects models to converge. However, we were able to reach convergence without taking this measure, and instead we log transformed both the response and explanatory variables. 

We agree that scaling and centering mass and length measurements is good practice for running mixed effects models. We repeated the analyses using the scaled variables, and the results did not qualitatively change. We offer both versions of the models in the R script for data analysis: https://doi.org/10.6084/m9.figshare.24996476.v2. 

We have deleted the sentence of concern and now specify that we scaled and log-transformed the variables. 

Reviewer #2:

“I enjoyed reading this manuscript. It’s well-written, and the motivation was well justified and explained, as were the methods. I particularly appreciated the explanations given for modelling choices throughout the methods section. I have no major concerns, only a minor comment:”

Response: We are pleased that the reviewer enjoyed the manuscript and found our motivations justified and modelling choices well explained.

“It seems the authors were careful to minimize the possibility that growth differences and temporal sampling contributed to major between-pond variation. I wonder if the authors looked at spatial autocorrelation among the sample sites. From the results it doesn’t seem obvious that there was spatial autocorrelation or potential for isolation by distance patterns in trait variation (with one site being markedly further away). But perhaps a lack of autocorrelation would also be interesting and support the idea that trait variation is more related to the pond-specific properties of salamander populations and environmental factors. In any case whether spatial autocorrelation was explored, or why the authors chose not to, should be noted in the Methods.”

Response: We agree that an analysis of spatial autocorrelation would be interesting even if no trend were found. However, given the small sample size of ponds in this study, we were concerned that any result of a positive/negative trend, or lack of a trend, would not be reliable. Our study aims did not specifically center around spatial autocorrelation and so our sampling design did not generate data sufficient for such analyses. Further, sampling enough ponds to reliably assess spatial autocorrelation, while minimizing temporal effects on morphology, would not have been feasible. We think the question of spatial autocorrelation is valuable though and would be a nice extension to this study’s report on between-pond variation. We will consider this in future projects that build off of this work. 

We have added a statement regarding this to our methods (Line 212): 

“Our sample size did not permit reliable examination into the role of spatial autocorrelation in explaining between-pond variation”.

---

## [Editor Report · Decision Letter 1]

6 Feb 2024

Trait variation in patchy landscapes: morphology of spotted salamanders (Ambystoma maculatum) varies more within ponds than between ponds

PONE-D-23-36372R1

Dear Dr. Daversa,

We’re pleased to inform you that your manuscript has been judged scientifically suitable for publication and will be formally accepted for publication once it meets all outstanding technical requirements. Congratulations!

Kind regards,

Dr. Janice L. Bossart

Academic Editor

PLOS ONE
---

## [Editor Report · Acceptance letter]

26 Mar 2024

PONE-D-23-36372R1 

PLOS ONE

Dear Dr. Daversa, 

I'm pleased to inform you that your manuscript has been deemed suitable for publication in PLOS ONE. Congratulations! Your manuscript is now being handed over to our production team.

Kind regards, 

on behalf of

Dr. Janice L. Bossart 

Academic Editor

PLOS ONE